# Assessing the Performance of a Dual-Speed Tool When Friction Stir Welding Cast Mg AZ91 with Wrought Al 6082

**DOI:** 10.3390/ma17153705

**Published:** 2024-07-26

**Authors:** Krzysztof Mroczka, Carter Hamilton, Aleksandra Węglowska, Mateusz Kopyściański, Stanisław Dymek, Adam Pietras

**Affiliations:** 1Department of Materials Engineering, Faculty of Materials Science and Physics, Cracow University of Technology, 31-155 Kraków, Poland; kmrocz@gmail.com; 2Department of Mechanical and Manufacturing Engineering, College of Engineering and Computing, Miami University, Oxford, OH 45056, USA; 3Łukasiewicz—Upper Silesian Institute of Technology, The Welding Centre, 44-100 Gliwice, Poland; aleksandra.weglowska@is.lukasiewicz.gov.pl (A.W.); adam.pietras@git.lukasiewicz.gov.pl (A.P.); 4Faculty of Metal Engineering and Industrial Computer Science, AGH University of Science and Technology, 30-059 Kraków, Poland; mateusz.kopyscianski@gmail.com (M.K.); gmdymek@cyfronet.pl (S.D.)

**Keywords:** dual-speed, friction stir welding, dissimilar metals, simulation, magnesium, aluminum

## Abstract

A novel dual-speed tool for which the shoulder and pin rotation speeds are separately established was utilized to friction stir weld cast magnesium AZ91 with wrought aluminum 6082-T6. To assess the performance and efficacy of the dual-speed tool, baseline dissimilar welds were also fabricated using a conventional FSW tool. Optical microscopy characterized the weld microstructures, and a numerical simulation enhanced the understanding of the temperature and material flow behaviors. For both tool types, regions of the welds contained significant amounts of the AZ91 primary eutectic phase, Al_12_Mg_17_, indicating that weld zone temperatures exceeded the solidus temperature of α-Mg (470 °C). Liquation, therefore, occurred during processing with subsequent eutectic formation upon cooling below the primary eutectic temperature (437 °C). The brittle character of the eutectic phase promoted cracking in the fusion zone, and the “process window” for quality welds was narrow. For the conventional tool, offsetting to the aluminum side (advancing side) mitigated eutectic formation and improved weld quality. For the dual-speed tool, experimental trials demonstrated that separate rotation speeds for the shoulder and pin could mitigate eutectic formation and produce quality welds without an offset at relatively higher weld speeds than the conventional tool. Exploration of various weld parameters coupled with the simulation identified the bounds of a process window based on the percentage of weld cross-section exceeding the eutectic temperature and on the material flow rate at the tool trailing edge. For the dual-speed tool, a minimum flow rate of 26.0 cm^3^/s and a maximum percentage of the weld cross-section above the eutectic temperature of 35% produced a defect-free weld.

## 1. Introduction

Since its development at TWI Cambridge, UK, in 1991, friction stir welding (FSW) has successfully joined numerous metallic material systems primarily utilizing a monolithic tool consisting of a shoulder and a pin rotating together at the same rate, i.e., a conventional tool, as well summarized in the review articles by Sambasivam et al. [1] and Grimm et al. [2]. Advancements in FSW technology, however, have explored various tool geometries, pin shapes, shoulder shapes, and orientations, such as the two-shouldered bobbin tool. Dumpala et al. [3] reviewed the development of the bobbin tool, and recently, Li et al. [4] effectively applied the bobbin tool design to joining magnesium, while Yadav et al. [5] coupled the bobbin tool with machine learning to weld aluminum. Contemporary developments in tool design now liberate the rotational movement of the pin from the shoulder. Initially, tools were developed for which the shoulder remained stationary as the pin rotated about the tool axis. For example, Barbini et al. [6] utilized such a “stationary shoulder” tool to join dissimilar aluminum alloys, and Sinhmar et al. [7] employed the stationary shoulder design to reduce and control heat input during the welding of aluminum in 2014. Similarly, Sundar et al. [8] found that the stationary shoulder reduced heat input and the width of the heat-affected zone when joining aluminum 6061, thereby improving microstructural characteristics and mechanical properties.

Another evolution in tool design is one that permits the rotation of the shoulder and the pin at different speeds, i.e., a dual-speed tool. The rotational speed of the shoulder primarily controls the heat input in the weld region, while the rotational speed of the pin principally influences the mixing within the process zone. To create favorable thermodynamic conditions for a high-quality weld, the rotational speed of the shoulder is typically set several times lower than that of the pin, especially when applied to dissimilar metal welds. A tool that would allow fully independent shoulder and pin rotation would be complicated and expensive, so a compromise solution is one that employs a planetary mechanism that allows the rotational speeds of the shoulder and pin to be unique but dependent on the gearing system, i.e., the gear ratio determines the pin rotation speed relative to that of the shoulder.

Aluminum 6082 is a precipitation-strengthened alloy that is age-hardened by the *GP* Zones → *β*″ (250 °C) → *β*′ (300 °C) → *β* (475 °C) precipitation sequence. Ultimately, the *β* phase dissolves at a solution heat treat temperature of 525 °C [9]. Typical process temperatures during friction stir welding of aluminum alloys often exceed the solution heat treat temperature(s) of the aluminum alloy(s), promoting *GP* zone precipitation upon cooling. If *GP* zones nucleate under these conditions, the hardness of the process zone recovers relative to the advancing/retreating sides and the thermo-mechanically affected zone. The chemical composition of Mg AZ91 is Al (~9%), Zn (~0.7%), and Mn (~0.2%), with the balance of Mg at ~90% [10,11]. Sen and Puri [12] noted the presence of two brittle intermetallic structures in the Mg–Al system: (1) the primary eutectic phase Al_12_Mg_17_ occurring at ~68% Mg/437 °C and (2) the secondary eutectic Al_3_Mg_2_ occurring at ~36% Mg/450 °C (the phase diagram for the Mg–Al system is also found in reference [13]). In AZ91, the solubility limit of Al in Mg is ~10%, corresponding to a solidus temperature of 470 °C, which is near the primary eutectic temperature; therefore, liquation in the AZ91 during FSW is a strong possibility. Should liquation occur, the formation of the brittle eutectic structures would ensue upon cooling, strongly influencing the mechanical integrity of the joint.

Several friction stir welding studies have employed conventional tools to join magnesium alloys with aluminum alloys and have evaluated the extent and impact of eutectic formation. For example, McLean et al. [14] joined AZ31 with Al 5053 and clearly identified the liquation of AZ31 within the weld zone. The liquation in the process zone produced the brittle Al_12_Mg_17_/α-Mg structure as a divorced lamellar eutectic. Joining AZ91 sheets with Al 6111 through friction stir spot welding, Gerlich et al. [15] determined that irrespective of the alloy orientation, i.e., alloy placement on top or bottom, welding temperatures exceeded the eutectic and solidus temperatures of AZ91 and the Al_12_Mg_17_ eutectic formed. Buffa et al. [16] considered AZ31/Al 6016 friction stir welds and identified the eutectic structure in the weld zone. Buffa, however, was able to produce defect-free welds with the AZ91 alloy oriented on the advancing side. Likewise, Sameer et al. [17], in their FSW study of AZ91 and Al 6082, demonstrated that defect-free joints could be produced despite the eutectic phase in the process zone if AZ91 were oriented on the advancing side. Through the application of friction stir processing to AZ91, Asadi et al. [18] concluded that most flow occurs on the advancing side, with some regression flow even occurring on the retreating side.

Friction stir welding research has failed to provide definitive guidelines regarding which alloy should be placed on which side of the weld (advancing or retreating) when joining dissimilar metals. Some studies have concluded that superior weld quality is achieved by placing the lower-hardness alloy on the advancing side and the higher-hardness alloy on the retreating side, while others have convincingly concluded the opposite. Consider the following: Park et al. [19] studied the joining of aluminum 5052-H32 with 6061-T6 and concluded that superior weld quality is achieved when 5052 (the softer alloy) is placed on the advancing side. Likewise, in their study of welding 6061-T6 and 7075-T6, Guo et al. [20] concluded that 6061 (the softer alloy) should be placed on the advancing side. However, Amancio-Filho et al. [21] concluded that 2024 (the harder alloy) should be placed on the advancing side when joining 6056-T4 and 2024-T3, and Reza-E-Rabby et al. [22] proposed that weld quality is optimized with 2050 (the harder alloy) on the advancing side when joining 6061-T6 and 2050-T4. In general, the advancing side of a friction stir weld is relatively hotter than the retreating side. Hamilton et al. [23], therefore, suggested that whichever workpiece has higher flow stress at processing temperatures should be placed on the advancing side, thereby augmenting plastic flow within the process zone of the alloy with the greater flow stress.

The above research efforts employed conventional tools without an offset of the tool towards either the advancing or retreating sides; however, an offset can play a crucial role in the ultimate weld quality. Using a conventional tool with a threaded pin and concave shoulder, Deng et al. [24] investigated the friction stir welding of AZ31 to Al 2024. Arranging AZ31 on the advancing side and Al 2024 on the retreating side, Deng studied the effect of tool offset on weld quality and determined that an offset to the aluminum alloy of 0.5 mm (retreating side) provided the highest weld quality. It was also discovered that offsetting the tool toward the aluminum reduces the formation of eutectic phases (both Al_12_Mg_17_ and Al_3_Mg_2_) [12]. Offsetting the aluminum decreased the amount of heat supplied to the AZ31 alloy and, therefore, reduced the extent of liquation in this alloy. Jadav et al. [25] considered the combination of AZ31 with Al 6061, and though these authors maintained the same weld configuration, i.e., Mg alloy on the advancing side, they concluded that moving the tool towards the Mg alloy provided the best weld quality, contrary to Deng’s work. Their study also considered cooling media and pin radii in conventional tools and concluded that the smallest pin radius (5 mm in their research) produced defect-free welds, no doubt because of the lower heat input associated with the smaller pin, which helped to minimize the liquation of the AZ31 alloy.

Whether from the configuration of the alloys, tool offset, or the process parameters and tool geometries (or all), the heat input clearly impacts the weld quality when joining Mg alloys with Al alloys. A balance exists between keeping the heat input low enough to minimize liquation in the Mg alloy and to mitigate brittle eutectic formation and then keeping the heat input high enough to promote sufficient plastic flow to create the joint. However, how to establish and/or define this balance is not clearly defined for the Mg/Al material system. For the FSW of Mg AZ91 with Al 6082, this study employs a dual-speed tool for which the pin and shoulder rotation speeds are separately established through a planetary gear system and compares its efficacy against welds produced with a conventional tool. Various rotation and welding speeds are considered for each tool, and offsets to both sides of the weld centerline are employed for the conventional tool, but no offset is used for the dual-speed tool. Ultimately, this study defines a “processing window” for the dual-speed tool that maximizes weld quality and performance, which is achieved at relatively higher weld speeds than the conventional tool.

## 2. Materials and Methods

Figure 1 displays the dimensions and schematics for the conventional tool (Figure 1a) and the dual-speed tool (Figure 1b) utilized in this investigation. The radius of the conventional tool shoulder is 14 mm with a 2.5 mm pitched scroll, and the radius of the dual-speed tool is 12 mm with the same pitched scroll. Workpieces of 100 × 250 × 6 mm dimensions of cast Al 6082-T6 and Mg AZ91 were procured for friction stir welding with the conventional tool, and workpieces of 100 × 250 × 4 mm dimensions were obtained for joining with the dual-speed tool. The Łukasiewicz—Upper Silesian Institute of Technology in Gliwice, Poland—performed all friction stir welding trials for this investigation. Prior to processing, any oxide layers were removed from the weld seam by sanding the plate edges and cleaning with solvent to remove contaminants.

For both tools, force control at 30 kN was applied during welding. Regarding the placement of the alloys on the advancing side (AS) or the retreating side (RS), following the reasoning presented in reference [23], the aluminum alloy, which has the higher flow stress at elevated temperatures, is placed on the AS and the magnesium alloy on the RS. Such an orientation should facilitate plastic flow in the aluminum alloy due to its placement on the relatively hotter AS. For the conventional tool, offsets to the aluminum (advancing) side and to the magnesium (retreating) side were considered. The authors’ results on joining AZ91 with Al 6082 utilizing a conventional tool without an offset are published elsewhere in reference [26]. For the dual-speed tool, welds were obtained with no offset to either the advancing or retreating sides and no tilt angle. The rotation and welding speeds were varied to identify the optimal settings for weld quality between these alloys. The process parameters employed are listed in Table 1—two unique conditions for the conventional tool and seven unique conditions for the dual-speed tool. After processing, welded panels cooled naturally to room temperature. For optical microscopy, welds were etched with a reagent containing C_6_H_2_(NO_2_)_3_OH 4.2 g, CH_3_COOH 1 mL, H_2_O 10 mL, and C_2_H_5_OH 96% 75 mL to highlight the Al_12_Mg_17_ intermetallic phase.

For this study, the authors’ previously developed simulation based on joining dissimilar aluminum alloys was adapted to this material system and process conditions [27,28]. The simulation links temperature and material flow and utilizes the temperature-dependent thermal properties for each alloy, i.e., thermal conductivity (*k*) and specific heat capacity (*c_p_*). For AZ91, the thermal conductivity and heat capacity are taken from Bannour et al. [29], and for Al 6082, they are taken from Gao et al. [30] and from Zahra et al. [31]. For each time step of the simulation, the temperature-dependent flow stress, *σ_e_*, and viscosity, *μ*, are determined based on the Sheppard–Wright formulation (Equation (1)) and the Zener–Hollomon parameter, *Z* (Equation (2), where $\dot{\varepsilon }$ is the strain rate) and their associated material constants, i.e., *Q*, *A*, α, and *n* [32].
(1)$$\sigma_{e}=\frac{1}{\alpha }sinh^{-1}\left[{\left(\frac{Z}{A}\right)}^{\frac{1}{n}}\right]$$
(2)$$Z=\dot{\varepsilon }exp\left(\frac{Q}{RT}\right)$$

For AZ91, these constants are taken from Raghunath et al. [33] and Wang et al. [34], and for Al 6082, they are taken from Wang et al. [35]. These values are presented in Table 2. The simulation utilizes a torque-based approach to determine heat input to the workpiece materials; therefore, the coefficient of friction, *μ*, is a critical material property in the numerical environment. The coefficient of friction data is taken from Chelliah et al. [36] and Srinivasan et al. [37] for AZ91 and from Threadgill et al. [38] for Al 6082. The current simulation utilizes a solution methodology consistent with the authors’ prior simulations, including boundary conditions for velocities around the tools, flow stress, viscosity, strain rate, temperature, and slip [27,28].

A representative solid model of the FSW process from the simulation (performed in Comsol) is shown in Figure 2a. To validate the simulation, workpieces of AZ91 were friction stir welded with sixteen k-type thermocouples embedded into the workpieces—eight thermocouples on each side of the weld. For each side of the weld, three thermocouples were embedded into the surface 16 mm from the weld centerline, two were embedded into the midplane of the weld cross-section 9 mm from the weld centerline, and three were embedded into the bottom plane 6 mm from the weld centerline. Figure 2b presents the measured and simulated temperature profiles as a function of time on the AS surface at 16 mm from the weld centerline, i.e., the profile shows the rise in temperature with the approaching tool and the decrease in temperature with the departing tool. The profiles highlight the agreement between the thermocouple data and the simulation temperatures, especially the peak temperature achieved at this thermocouple location; moreover, this agreement is representative of the correlations at the other thermocouple locations. The simulation slightly overpredicts the increasing temperature rate and slightly underpredicts the falling temperature rate. However, given the agreement in maximum temperatures, which ultimately determine the phases present in the weld microstructure, it is reasonable to utilize the temperature distributions predicted by the simulation to represent the temperature history in the weld zone and to provide reliable analysis tools.

## 3. Results and Discussion

### 3.1. Conventional Tool

In the referenced previous study, the authors joined Mg AZ91 (retreating side) and Al 6082 (advancing side) with a conventional tool (the same tool as shown in Figure 1a) and no offset [26]. The resulting process zone was comprised of interleaved layers of aluminum and the Al_12_Mg_17_/α-Mg eutectic structure, consistent with the observations of Mclean [14], Gerlich [15], Buffa [16], and Sameer [17], as mentioned earlier. Further examination of the eutectic layers by SEM and EDS characterization, however, revealed that the eutectic bands and surrounding matrix were, in fact, a distinct mix of the major phases from the Al-Mg phase diagram. EDS spectra revealed four primary components of the weld zone: (a) α-Mg solid solution near the solubility limit of Al in Mg; (b) primary Al_12_Mg_17_/α-Mg eutectic; (c) Al 6082; and (d) secondary Al_3_Mg_2_ eutectic. Xu et al. [39] observed this secondary eutectic in the Mg/Al system, which occurs at ~36% Mg and ~64% Al, during investigations of Mg AZ31 and Al 5A06 welds.

In the current study, tool offsets to AS (aluminum) and RS (magnesium) were considered. The weld microstructures in Figure 3 were all acquired at 710 RPM tool rotation and 90 mm/min weld speed [26]. The overall weld shape for which the root is wider than the surface aligns with the observations of Mehta et al. [40] in their study of AZ31B/Al6061 friction stir welds in which “bulbous” shape welds were produced. As they noted, the flow of the aluminum alloy dominates the upper weld region, producing a trapezoidal shape in this area, while strong mixing of the alloys toward the root produces a wider weld region. The work by Chen et al. [41], which investigated friction stir welding of the same alloys as Mehta et al., also substantiates this weld shape when they observed that strong vertical flow along the pin is reflected by the weld bottom, producing significant interleaving of the alloys in this region.

In Figure 3a,c, the tool is offset ~0.6 mm towards the aluminum alloy, and in Figure 3b,d, the tool is offset ~0.5 mm towards the magnesium alloy. With the tool offset toward the aluminum, the amount of intermetallic phase, primarily eutectic, is relatively less, and the distribution of eutectic bands and areas related to the AZ91 magnesium alloy indicate a structurally symmetrical joint. This confirms the previously mentioned conclusions from Deng [24] on obtaining a high-quality joint under such FSW conditions. When the tool is offset toward the magnesium, a more significant share of the Mg alloy is observed at the weld face and on the advancing side, with more aluminum alloy consequently transferred to the retreating side, creating an asymmetrical nugget zone of the workpiece materials.

During FSW, material is extruded from the advancing side surface into the weld zone. If surface temperatures exceed the solidus temperature (470 °C), then liquation will occur, and the subsequent liquid will be forced into the process zone and interwoven with the solid-state workpiece material. Upon cooling and liquid solidification, the Al_12_Mg_17_ (437 °C eutectic temperature) and/or Al_3_Mg_2_ (450 °C eutectic temperature) eutectic structures would form. Figure 4 presents the temperature distributions for no offset, AS offset, and RS offset at 710 RPM/90 mm/min determined from the simulation on a reference plane across the weld zone located 6 mm behind the tool [26]. Also indicated in the figure are isotherms representing 470 °C (the solidus temperature of AZ91) and 437 °C (the eutectic temperature of AZ91). The previous work by the authors without offset demonstrated that despite the eutectic structure present in the weld zone, a defect-free weld could be obtained at 710 RPM/90 mm/min. Here, for an offset to the AS (aluminum side), the 470 °C isotherm shifts downward on the advancing side, indicating that a larger area of the AS lies above this temperature relative to the no offset condition. The temperature profile on the retreating side, however, remains relatively unchanged, though a slight downward shift in the 470 °C isotherm is noted, along with a small shift to the right in the 437 °C isotherm. As a result, the higher temperatures enhance flow conditions in the Al 6082 while also promoting a subtle rise in the amount of eutectic from AZ91. The symmetric nugget and eutectic bands in Figure 3a underscore this effect.

When the tool is offset to the RS (magnesium side), the temperature distribution effectively matches that of the no-offset profile. Under these conditions, therefore, the flow of the aluminum is essentially unchanged, and the amount of eutectic coming from the AZ91 also remains relatively the same. However, due to the offset toward the retreating side, the tool sweeps relatively more magnesium toward the advancing side. With similar flow behavior in Al 6082, the resulting nugget is asymmetric, with a concentration of the eutectic bands toward the surface of the workpieces on the advancing side, as evidenced in Figure 3b. Also, as the AZ91 flows into the weld zone on the AS, it does not fully interleave with other stir zone material, forming the void, also seen in Figure 3b, and forcing the stir zone material deeper into the cross-section and to the retreating side. Thus, based on these results, higher weld quality is obtained by an offset of the conventional tool to the aluminum side due to the enhanced flow without a significant increase in eutectic formation. An offset to the magnesium side did not necessarily produce more eutectic structure, but the relative decrease in the flow behavior of Al 6082 promoted an asymmetric nugget with eutectic agglomeration toward the top of the zone.

### 3.2. Dual-Speed Tool

As the conventional tool results suggest, reducing heat input and keeping surface temperatures below critical temperatures could constrain eutectic formation. Heat input, however, cannot be kept too low when considering the aluminum workpiece. With Al 6082’s solution heat treatment temperature at 525 °C, welding temperatures significantly less than this value could impede satisfactory mixing of the aluminum workpiece compared to the mixing that is achieved at higher process temperatures. Herein, however, lies the potential advantage of the dual-speed tool. With the ability to establish different pin and shoulder rotation speeds, the dual-speed tool affords greater flexibility over the heat input and material flow without an offset. Such flexibility subsequently creates the opportunity for higher weld speeds than the conventional tool with comparable or superior weld quality.

As presented in Table 1, seven combinations of pin rotation, shoulder rotation, and weld speeds were explored for the dual-speed tool, with weld speeds ranging from 140 mm/min up to 450 mm/min. Of the seven combinations, only the pin rotation of 1800 RPM, shoulder rotation of 450 RPM, and weld speed of 450 mm/mm (1800/450/450) produced a defect-free weld (the weld cross-section is shown in Figure 5a). All other process parameter combinations displayed cracks in the fusion zone, as representatively shown in Figure 5b for the 1800/450/280 parameter set. Here, a crack appears along an interlayer boundary between the aluminum and the magnesium eutectic on the AS of the weld.

To elucidate the correlation between process parameters and weld quality, the temperature distributions predicted by the simulation on a reference plane (extending 20 mm from the weld centerline into both the AS and RS with a height equal to the workpiece thickness, i.e., 40 × 4 mm) 6 mm behind the pin were considered. These distributions are shown in Figure 6 for each of the seven process parameter combinations. The images are ordered according to the percentage of the cross-sectional area above the primary eutectic temperature of AZ91, 437 °C. Thus, the “hottest” weld, i.e., the weld achieving the highest percentage of the cross-sectional area above 437 °C, is at the top of the figure, with the “coldest” weld at the bottom of the figure. These percentages are shown in Table 3. It must be noted, of course, that these values depend on the size of the reference plane selected. For the dual-speed tool, the shoulder radius is 13.5 mm; therefore, the reference plane extends 6.5 mm into the advancing and retreating sides beyond the tool’s shoulder. The slowest weld speed for the dual-speed tool produced the highest heat input, and it is not surprising, therefore, that 75% of the planar cross-section is above 437 °C for the 1800/450/140 parameter set. As such, within the stir zone where mixing occurs, a significant amount of brittle Al_12_Mg_17_/α-Mg eutectic is produced under these conditions, which then promotes cracking in the fusion zone. As the percentage of the planar cross-section above the eutectic temperature decreases, the amount of brittle eutectic in the fusion zone also decreases, lowering the chances of weld zone cracks. One can infer, therefore, that a threshold value of the planar cross-section above *T_eutectic_* (as a measure of the heat input) exists above which a defect-free weld is unlikely due to the extent of and cracking of the eutectic structure but below which a defect-free weld is achievable. Indeed, at a cross-sectional percentage of 31%, a defect-free weld is produced under the 1800/450/450 conditions (italicized/bolded in Table 3 for emphasis).

However, as previously discussed, heat input cannot be considered the lone driver of weld quality for these dissimilar joints. Though the lower heat input helps mitigate the influence of the brittle eutectics on weld performance, temperatures must also be sufficient to promote adequate material flow, especially for the Al 6082, to form a defect-free weld. The “coldest” weld for the dual-speed tool was produced under the 1400/350/450 combination with only 11% of the planar cross-section above *T_eutectic_*. As shown in Figure 7a, however, despite a relatively small amount of the eutectic structure (the optically blue interlayers in the weld zone), the weld zone shows inadequate mixing to achieve a consolidated weld. To assign a metric to the material flow, the reference plane shown in Figure 7b was defined at the trailing edge of the tool in the simulation, and the flow rate (in cm^3^/s) was determined. These flow rate values are also presented in Table 3. Though the percentage of the cross-section above *T_eutectic_* is only 11% for the 1400/350/450 parameter combination, the flow rate through the trailing edge reference plane is only 20.5 cm^3^/s compared to the 26.2 cm^3^/s obtained for the optimal 1800/450/450 parameter set. Again, one may infer that a threshold value of flow rate exists below which material flow is insufficient to produce a defect-free weld but above which the flow rate is sufficient to obtain a quality weld. Based upon these results, the recommended “processing window” required to produce a defect-free weld for the Mg AZ91-Al 6082 material system with this dual-speed tool would be a minimum flow rate of 26.0 cm^3^/s and a maximum percentage of the weld cross-section above the eutectic temperature of 35%.

These thresholds certainly could or would be different for various factors such as tool geometry, material systems, workpiece dimensions, etc. And, as previously mentioned, the percent cross-section and flow rate values derive from how the reference planes are defined. The broader outcome of this study demonstrates that a dual-speed tool without offset can achieve comparable or superior weld quality to a conventional tool by providing greater flexibility over temperature and flow. The percent cross-section and flow rate values presented in Table 3 underscore that definable target values for weld quality exist for this material system and demonstrate that the dual-speed tool gives more flexibility in achieving these targets. Though the range of successful process parameters shown here is narrow, achieving a defect-free weld with a dual-speed tool and no offset is possible for AZ91-Al 6082 dissimilar welds, even with the Al alloy placed on the advancing side. The weld speed used to produce a defect-free weld with the dual-speed tool is noticeably higher than that of the conventional tool used in the previous study, 450 mm/min compared to 90 mm/min. Such higher weld speeds present practical advantages in manufacturing in terms of throughput and efficiency.

## 4. Conclusions

Mg AZ91 was joined with Al 6082-T6 through friction stir welding utilizing a novel dual-speed tool with no offset and a conventional tool with offsets to the advancing and retreating sides. The primary goal of this study was to demonstrate the efficacy of the dual-speed tool to join Mg/Al dissimilar metals and to assess its performance against the conventional tool. A numerical simulation of the process provided a better comprehension of the temperature profiles through the weld zones and helped formulate the following conclusions:Defect-free friction stir welds were produced with the dual-speed tool and no offset at relatively higher weld speeds than the conventional tool. With separate pin and shoulder rotation speeds, the dual-speed tool provides greater flexibility over material flow rate and temperature, thus helping to control eutectic formation. A minimum flow rate of 26.0 cm^3^/s and a maximum percentage of the weld cross-section above the eutectic temperature of 35% produced a defect-free weld with the dual-speed tool.Offsetting the conventional tool to the aluminum (advancing) side produced a higher-quality weld than offsetting to the magnesium (retreating) side. The offset to the aluminum side produced relatively less brittle and eutectic material in the weld zone and provided more uniform flow. The weld quality of the conventional tool was comparable to the weld quality of the dual-speed tool, but at lower weld speeds.For both tools, the weld surfaces reach temperatures equivalent to or in excess of the solidus temperature of Mg AZ91, leading to the liquation and extrusion of the liquid into the weld zone. As the liquid solidifies, the Al_12_Mg_17_/α-Mg eutectic structure forms (and to a lesser extent, the Al_3_Mg_2_/α-Al structure) in the weld zone as interleaved layers of workpiece materials.

## Figures and Tables

**Figure 1 materials-17-03705-f001:**
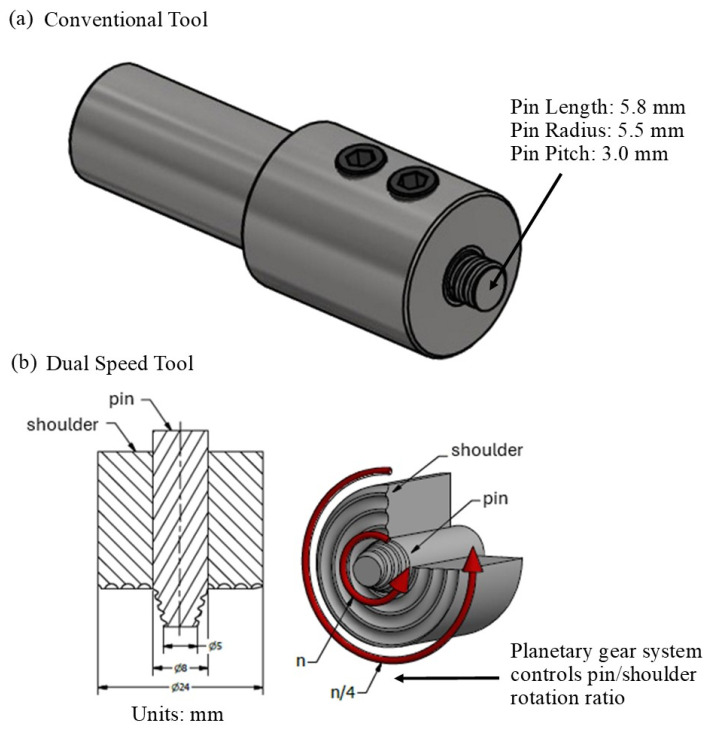
Schematic representations of (**a**) the conventional FSW tool and (**b**) the dual-speed FSW tool employed for this investigation.

**Figure 2 materials-17-03705-f002:**
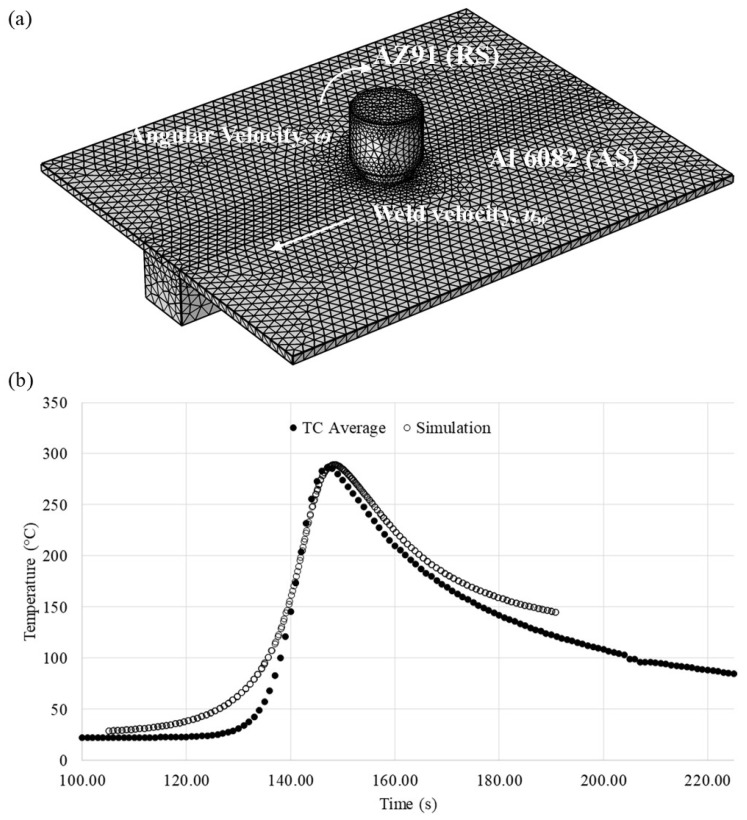
(**a**) Representative solid model of the FSW process taken from the simulation; and (**b**) representative comparison of measured and simulated temperature profiles located at 16 mm from the weld centerline on the AS (conventional tool shown).

**Figure 3 materials-17-03705-f003:**
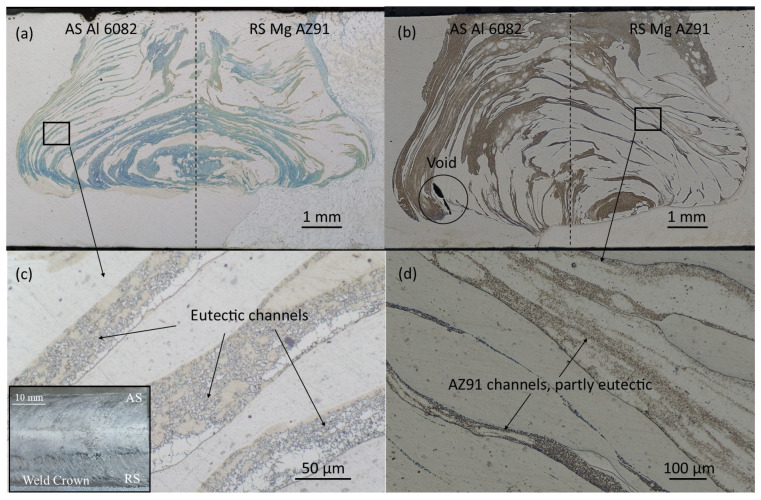
Weld microstructure at 710 RPM/90 mm/min: (**a**,**c**) ~0.6 mm offset towards aluminum alloy, (**b**,**d**) ~0.5 mm offset towards magnesium. Images (**c**,**d**) show higher magnifications of the eutectic channel layers (weld crown shown as inset to (**c**)) [26].

**Figure 4 materials-17-03705-f004:**
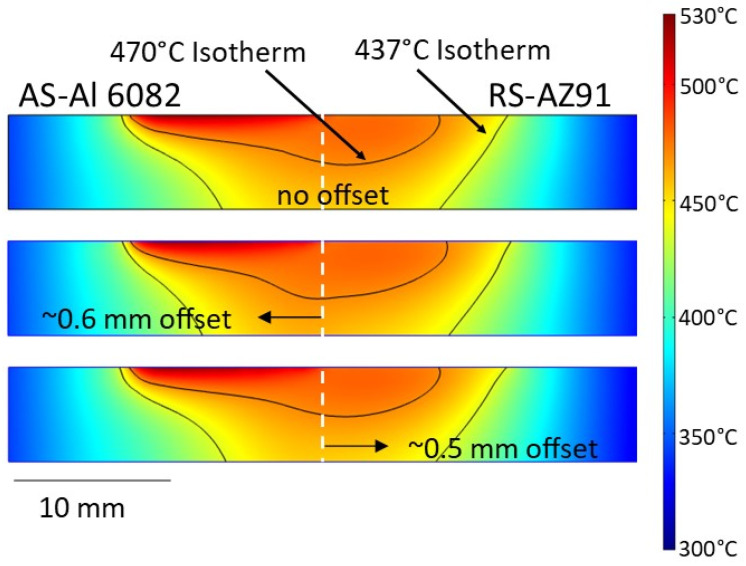
Simulation-based temperature distributions at 710 RPM/90 mm/min on a reference plane 6 mm behind the conventional tool for no offset, ~0.6 mm offset to Al 6082, and ~0.5 mm offset to AZ91 [26].

**Figure 5 materials-17-03705-f005:**
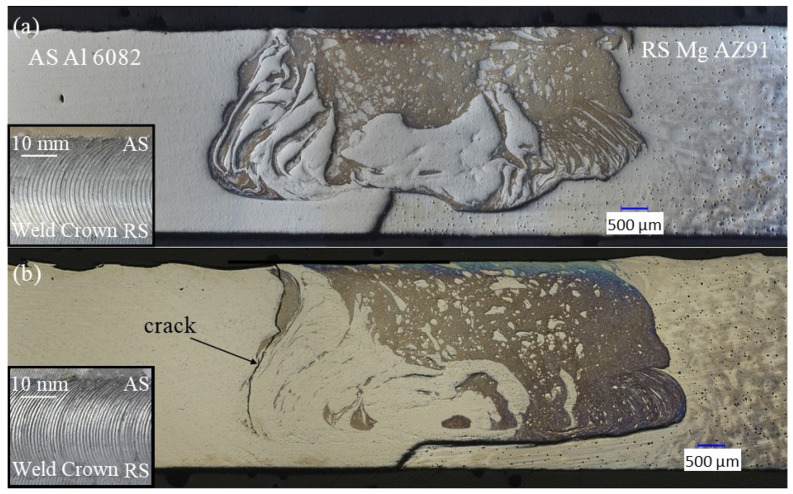
Microstructures produced with the dual-speed tool: (**a**) defect-free weld for the 1800/450/450 combination and (**b**) representative cracked weld for the 1800/450/280 combination. Images of weld crowns are shown as insets in each image.

**Figure 6 materials-17-03705-f006:**
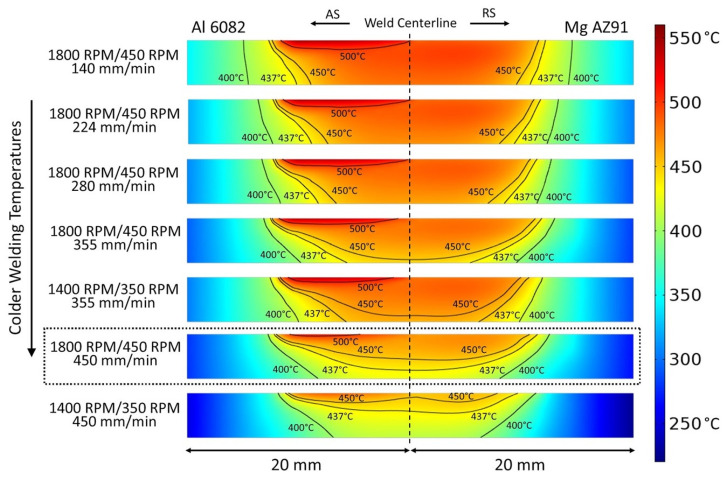
Temperature distributions from the simulations on a reference plane located 6 mm behind the tool pin.

**Figure 7 materials-17-03705-f007:**
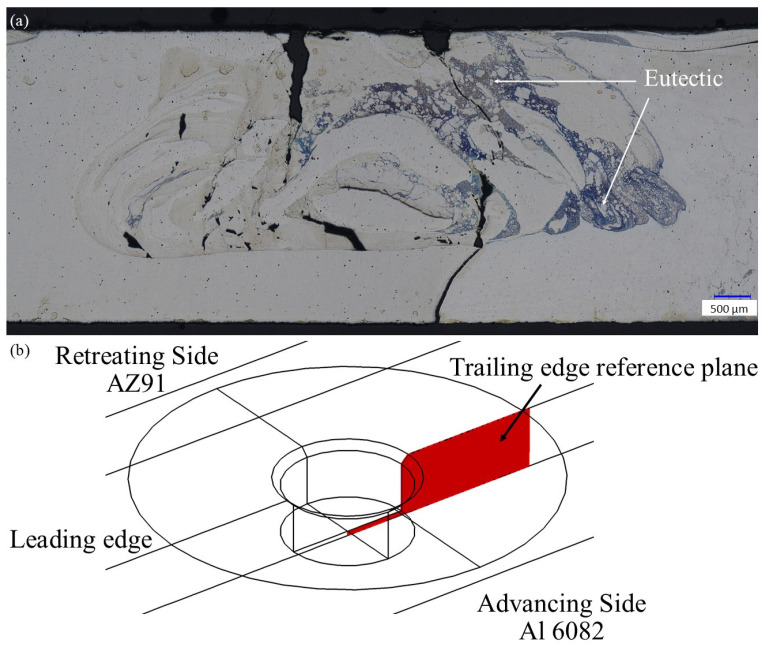
(**a**) Weld cross-section for the 1400/350/450 combination and (**b**) trailing edge reference plane defined in the simulation for material flow rate.

**Table 1 materials-17-03705-t001:** FSW process parameters employed for each tool type.

**Conventional Tool**		
**Shoulder Rotation (RPM)**		**Weld Speed (mm/min)**
710		90
710		140
**Dual-Speed Tool**		
**Shoulder Rotation (RPM)**	**Pin Rotation (RPM)**	**Weld Speed (mm/min)**
350	1400	350
350	1400	450
450	1800	140
450	1800	224
450	1800	280
450	1800	350
450	1800	450

**Table 2 materials-17-03705-t002:** Material constants for the Sheppard–Wright and the Zener–Hollomon formulations.

Alloy	*Q* (J/mol)	*A* (1/s)	*α* (1/MPa)	*N*
AZ91	177,500	2.8405 × 10^12^	0.021	5.578
Al 6082	168,000	3.0197 × 10^11^	0.024	4.709

**Table 3 materials-17-03705-t003:** Percent of cross-section of temperature reference plane above the AZ91 eutectic temperature and material flow rate through the trailing edge reference plane.

Parameter Set Pin/Shoulder/Weld Speed (RPM-RPM-mm/min)	%Cross-Section >*T_eutectic_*	Flow Rate at Trailing Edge (cm^3^/s)
1800/450/140	75%	26.7
1800/450/224	68%	26.6
1800/450/280	64%	26.5
1800/450/355	59%	26.4
1400/350/355	47%	20.5
** *1800/450/450* **	** *31%* **	** *26.2* **
1400/350/450	11%	20.5

## Data Availability

The raw data supporting the conclusions of this article will be made available by the authors upon request.

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
