# Peer review of "Assessing the Performance of a Dual-Speed Tool When Friction Stir Welding Cast Mg AZ91 with Wrought Al 6082"

_materials, 2024, doi:10.3390/ma17153705_

Round 1

Reviewer 1 Report

Comments and Suggestions for Authors

The authors joined Mg AZ91 and Al 6082-T6 through friction stir welding utilizing a novel dual-speed tool with no offset and a conventional tool with offsets to the advancing and retreating sides. The goal is to demonstrate the efficacy of the dual-speed tool to join Mg/Al  and to assess the performance versus the conventional tool. A numerical simulation of the process is performed to highlight the temperature profiles through the weld zones.

The references and the state-of-the-art presentation cover the relevant results in the field. The drawbacks of the published research are stressed out in the paper.

Paper contributions are highlighted in relation to the authors’ previous research (references 23, 26-28). The paper is based on the results presented in paper 26.

The results cover both the use of a conventional tool and dual-speed tool.

Author Response

Thank you to the reviewers and the editor for considering our manuscript and for the thoughtful comments and suggestions to improve its clarity, rigor, and quality. The following are our responses to these comments, and the text has been revised as appropriate to reflect these changes. Revisions associated with comments from Reviewer 2 are highlighted in yellow in the revised manuscript. Overlapping comments from Reviewers 2 and 3 are highlighted in grey. The comments and suggestions from the reviewers have strengthened the manuscript, and we appreciate their time and effort.

Reviewer 1

Comment 1. The authors joined Mg AZ91 and Al 6082-T6 through friction stir welding utilizing a novel dual-speed tool with no offset and a conventional tool with offsets to the advancing and retreating sides. The goal is to demonstrate the efficacy of the dual-speed tool to join Mg/Al and to assess the performance versus the conventional tool. A numerical simulation of the process is performed to highlight the temperature profiles through the weld zones.

The references and the state-of-the-art presentation cover the relevant results in the field. The drawbacks of the published research are stressed out in the paper.

Paper contributions are highlighted in relation to the authors’ previous research (references 23, 26-28). The paper is based on the results presented in paper 26.

The results cover both the use of a conventional tool and dual-speed tool.

Response: Thank you to the reviewer for reading our manuscript and providing his/her assessment and opinion. We greatly appreciate the reviewer’s time and thoughtfulness.

Reviewer 2 Report

Comments and Suggestions for Authors

The following are my comments for this work:

It is an interesting work related to friction stir welding of AZ31 alloy with Al 6082 T6 alloy.

The abstract resumes quite well the most important aspects of the work.

Introduction: I recommend in figure 1 to  show the shoulder, the scroll and the pin of a typical FSW process.

Materials and methods are well described. Only some doubts:

Why authors use different thicknesses? (4 and 6 mm)

Table 1. Why authors do not test the combinations (shoulder rotation/pin rotation/weld speed):

350/1400/140

350/1400/224

350/1400/280?

About Sheppard-Wright and Zener-Hollomon models,  it is necessary to write those equations in the paper.

Results and discussions are analyzed in details. Some minor doubts must be explained. See the following questions:

Figure 4: How the authors measure the "offset"? Could the authors explain this?

Line 301: Why the author say: "the dual-speed tool affords greater control over the heat input..." Why is the reason for this?

Figure 5. Why 1800/450/280 generates cracks, but 1800/450/450 does not generate? The cooling rate, was the same in both cases?

I have a doubt related to cooling rate. Are there any control over this variable?

Conclusions

The conclusions are pertinent and they are well connected with experimental results.

Author Response

Thank you to the reviewers and the editor for considering our manuscript and for the thoughtful comments and suggestions to improve its clarity, rigor, and quality. The following are our responses to these comments, and the text has been revised as appropriate to reflect these changes. Revisions associated with comments from Reviewer 2 are highlighted in yellow in the revised manuscript. Overlapping comments from Reviewers 2 and 3 are highlighted in grey. The comments and suggestions from the reviewers have strengthened the manuscript, and we appreciate their time and effort.

Reviewer 2

Comment 1. It is an interesting work related to friction stir welding of AZ31 alloy with Al 6082 T6 alloy. The abstract resumes quite well the most important aspects of the work.

Response: Thank you to the reviewer taking the time to read our manuscript and for providing feedback. Also, the complimentary tone toward our paper is very much appreciated. Thank you for your help in improving its quality.

Comment 2. Introduction: I recommend in figure 1 to show the shoulder, the scroll and the pin of a typical FSW process.

Response: Per the suggestion of the reviewer, Figure 1 has been amended to show more detail of the pin and shoulder of the dual-speed tool. The revised Figure 1 now appears in the revised text.

Comment 3. Why authors use different thicknesses? (4 and 6 mm)

Response: The difference in workpiece thickness is simply due to the difference in the pin length between the two tool designs. For the conventional tool, the pin length was 5.8 mm (97% of the workpiece thickness) and for the dual-speed tool, the pin length was 3.7 mm (93% of the workpiece thickness). These values are typical to the tool designs used for the conventional and dual-speed tools in our studies.

Comment 4. Table 1. Why authors do not test the combinations (shoulder rotation/pin rotation/weld speed): 350/1400/140, 350/1400/224, 350/1400/280?

Response: Thank you for your question, and as you suggest, a wide range of parameters should be utilized to evaluate the weld quality. Ultimately in this study, the tests were carried out using the following parameters: 1400/350/...: 224, 355, 450 mm/min,and 1800/450/...: 140, 280, 355, 450 mm/min. Additional parameters, e.g. 1400/350/224, were actually also considered, but such manufactured joints completely failed when attempts were made to excise samples from them. In these instances, cracks typically appeared in similar locations as shown in Fig. 4a for 1400/350/280 sample. Therefore, out of the many experiments performed, seven parameter sets resulted in macroscopically sound joints, i.e., no defects were observed on the face and ridge. These variants were selected to develop the numerical model and present the results in this manuscript.

Comment 5: About Sheppard-Wright and Zener-Hollomon models, it is necessary to write those equations in the paper.

Response: The referenced equations have been added to the paragraph below Table 1 and the text appropriately revised.

Comment 6. Figure 4: How the authors measure the "offset"? Could the authors explain this?

Response: Prior to welding, the FSW tool offset was initially established at the beginning side of the fixed workpieces by caliper measurement. The tool was then traversed in the weld direction to the ending side of the workpieces, and the offset was again measured to verify that its value matched that as at the beginning. Due to the relatively short length of the welds and straight edges, the offset in the middle part was the same as at the beginning and end. Undoubtedly, in the case of long welds, the offset should be controlled automatically.

Comment 7. Line 301: Why the author say: "the dual-speed tool affords greater control over the heat input..." Why is the reason for this?

Response: In the first and second paragraphs of the Introduction, we discuss research studies available in the literature that demonstrate that the tool shoulder primarily controls heat input into the weld and that the pin primarily controls mixing within the stir zone (for example, references 6, 7 and 8). The dual-speed tool allows the rotation speed of the shoulder and the pin to be separately established, which we deemed as giving “greater control” over the heat input by setting a shoulder rotation speed that is unique from the pin rotation speed. In reflecting on the criticism of the reviewer, perhaps “greater flexibility” is a more appropriate term. As such, the references to “greater control” or “control” when speaking of the dual-speed tool have been amended to “greater flexibility” or “flexibility”. These amendments are highlighted in yellow in the revised text.

Comment 8. Figure 5. Why 1800/450/280 generates cracks, but 1800/450/450 does not generate? The cooling rate, was the same in both cases?

Response: Following welding, all welded panels naturally cooled in a room temperature environment. As such, all cooling rates were effectively identical; however, no specific efforts were made to control the cooling rates. For clarity, the following statement was added to the second paragraph of the Materials and Methods:

After processing, welded panels cooled naturally to room temperature.

Comment 9. I have a doubt related to cooling rate. Are there any control over this variable?

Response: As mentioned in the response to Comment 8, no specific efforts were made to control the cooling rates. The authors acknowledge that controlled and/or forced cooling rates can have a significant influence on the precipitation behavior following friction stir welding. Given that the goal of this research was to assess the efficacy of a dual-speed tool in relation to a conventional tool, the authors felt that it is was better not to introduce the cooling rate as another process parameter. As such, and as in the authors’ previous FSW studies, the welded panels were allowed to naturally cool to room temperature.

Comment 10. The conclusions are pertinent and they are well connected with experimental results.

Response: Again, thank you to the reviewer for reading our manuscript and giving his/her assessment.

Reviewer 3 Report

Comments and Suggestions for Authors

Kudos to the authors of the displayed article. Namely, the authors deal with the analysis of friction welding of two different metals. Aluminum and Magnesium with their impurities. Research is focused on a conventional friction welding tool and a new one that has dual rotation capability in terms of rotation speed. The advantages in terms of the defective layer are presented, where different parameters of rotation and displacement of the tool are analyzed. The center path of the tool has also been moved, towards aluminum or magnesium. Conclusions were made regarding the recommended mode, which gives the smallest joint defect (1800/450/450) as well as a shift in favor of aluminum, from the aspect of the appearance of the eutectic phase. The entire research is threatened by simulations regarding the resulting temperature and its effect on the quality of the joint.
My only complaint to the authors is that they should have presented the new tool in more detail. Accordingly, I am asking them to post another picture of the tool with double the rotation speed.

Author Response

Thank you to the reviewers and the editor for considering our manuscript and for the thoughtful comments and suggestions to improve its clarity, rigor, and quality. The following are our responses to these comments, and the text has been revised as appropriate to reflect these changes. Revisions associated with comments from Reviewer 2 are highlighted in yellow in the revised manuscript. Overlapping comments from Reviewers 2 and 3 are highlighted in grey. The comments and suggestions from the reviewers have strengthened the manuscript, and we appreciate their time and effort.

Reviewer 3

Comment 1. Kudos to the authors of the displayed article. Namely, the authors deal with the analysis of friction welding of two different metals. Aluminum and Magnesium with their impurities. Research is focused on a conventional friction welding tool and a new one that has dual rotation capability in terms of rotation speed. The advantages in terms of the defective layer are presented, where different parameters of rotation and displacement of the tool are analyzed. The center path of the tool has also been moved, towards aluminum or magnesium. Conclusions were made regarding the recommended mode, which gives the smallest joint defect (1800/450/450) as well as a shift in favor of aluminum, from the aspect of the appearance of the eutectic phase. The entire research is threatened [sic] by simulations regarding the resulting temperature and its effect on the quality of the joint.

Response: Thank you to the reviewer for reading our manuscript and providing his/her assessment and opinion. The positive tone and compliment toward our paper is very much appreciated. Thank you for your time and effort.

Comment 2. My only complaint to the authors is that they should have presented the new tool in more detail. Accordingly, I am asking them to post another picture of the tool with double the rotation speed.

Response: Per the suggestion of the reviewer, Figure 1 has been amended to show more detail of the pin and shoulder of the dual-speed tool. The revised Figure 1 now appears in the revised text.

Round 2

Reviewer 2 Report

Comments and Suggestions for Authors

All my suggestions were taken into account. In the second version of the paper, figures are more clear and completes.